# Contamination Assessment of Mangrove Ecosystems in the Red Sea Coast by Polycyclic Aromatic Hydrocarbons

**DOI:** 10.3390/ijerph19095474

**Published:** 2022-04-30

**Authors:** Abdulrahman H. Alhudhodi, Abdulilah K. Alduwais, Zaid M. Aldhafeeri, Mohammed Ahmad S. Al-Shamsi, Badr H. Alharbi

**Affiliations:** National Centre for Environmental Technology (NCET), Life Science & Environment Research Institute (LSERI), King Abdulaziz City for Science & Technology (KACST), Riyadh 12354, Saudi Arabia; aalhadadi@kacst.edu.sa (A.H.A.); aaldowas@kacst.edu.sa (A.K.A.); zaldhafeeri@kacst.edu.sa (Z.M.A.); mshamsi@kacst.edu.sa (M.A.S.A.-S.)

**Keywords:** PAH, mangrove, contamination, sediments, sources, Rabigh

## Abstract

Mangroves are known as a naturally based solution for climate mitigation and adaptation. Mangroves are at a potential risk of degradation by contaminants such as polycyclic aromatic hydrocarbons (PAHs). In this study, sixteen priority PAHs were analyzed and characterized in forty samples of mangrove seawater and mangrove sediments collected from two coastal areas (i.e., Sharm and Khor Rabigh) along the Red Sea Coast of Rabigh city in August 2013. We found that the average concentration of total PAH in mangrove sediments in the Sharam area (22.09 ng/kg) was higher than that in the Alkhor area (6.51 ng/kg). However, the average concentration of the total PAH in the mangrove seawater in the Alkhor area (9.19 ng/L) was double that in the Sharam area (4.33 ng/L). Phenanthrene and pyrene were the major components in both the mangrove seawater and sediment in all the investigated areas. We observed that the abundance of PAHs with 2–3 aromatic rings was dominant in sediment samples collected from both study areas. This abundance was also observed in seawater from the Sharam area. However, seawater samples from the Alkhor area had abundant PAHs with four aromatic rings. The majority of PAHs in sediment samples of both study areas originated from petrogenic sources, whereas the majority of PAHs in seawater samples originated from pyrogenic sources.

## 1. Introduction

Globally, approximately 50% of mangroves have been lost over recent decades. The main human activities for losing mangroves are reclamation, farming, aquaculture, deforestation, and urban development [1,2,3,4], with an estimate of 62% of global losses between 2000 and 2016 due to land-use change [5]. In coastal areas, mangrove ecosystems, in addition to being rich breeding and nursery grounds for various aquatic organisms, also act as a buffer between sea waves and the land and help stabilize the coastline and prevent coastal erosion [6]. Typically, these sites of mangroves are rich in debris and organic carbon and sheltered from strong wind and sea waves; therefore, they are a suitable environment for the deposition and buildup of contaminants with a slow rate of degradation under anoxic conditions [7,8]. Contaminants released from anthropogenic activities pose a threat to polluting mangrove waters and sediments, particularly polycyclic aromatic hydrocarbons (PAHs) [9,10].

PAHs are aromatic species with at least two fused benzene rings. They are prevalent in the environment due to their numerous sources. Because of their toxicity, carcinogenicity and mutagenicity, the United States Environmental Protection Agency (USEPA) has identified sixteen PAHs as priority pollutants. Typically, anthropogenic PAHs in marine environments are caused by industrial discharge, urban runoff, wastewater, engine oil, oil spills, ship and boat activities, and atmospheric deposition of industrial and traffic emissions [11,12]. PAH toxicity and risk assessment in aquatic sediment have been reported in many studies [13,14,15,16,17]. When PAHs are released to the environment as petroleum or petroleum products, the source is called petrogenic, whereas if PAHs are released as a result of inefficient or incomplete combustion, the source is called pyrogenic [18,19]. The identification of PAH sources is important for assessing sediment contamination. PAHs from petrogenic sources are more toxic than PAHs from pyrogenic sources [20,21,22].

Rabigh, a coastal city in Saudi Arabia located on the Red Sea, has experienced high industrial growth rates over the past three decades, with several heavy and light industries within the city boundaries, including oil refineries, petrochemical companies, cement plants, and power and desalination plants. Furthermore, King Abdullah Economic City and King Abdullah Port are approximately 25 km from the study area. The aim of this article is to investigate the PAH contamination level in mangrove ecosystems (i.e., sediments and seawaters) in the two selected study areas on the eastern coast of the Red Sea.

## 2. Materials and Methods

### 2.1. Study Area

Two areas of mangroves located along the Red Sea Coast close to the city of Rabigh, Saudi Arabia, were selected for the study, as shown in Figure 1. These two coastal areas (i.e., the Sharam area and Alkhor area) are rich in Avicennia marina and Rhizophora mucronata mangrove trees. The Sharam area regularly experiences light petrochemical and oil shipping activities of approximately 20–25 vessels a month. The mangrove in the Sharam area is located approximately 2 km north of a petrochemical company. However, the Alkhor area experiences various potential sources of contamination, including wastewater discharge, boat activities, waste oil spilled during oil changes and disposal, and to a lesser extent, fuel spilled during refueling operations, particularly in the small marina and the station of the border guards near the intersection of the open sea and the water body of Alkhor.

### 2.2. Meteorology Measurements

These meteorological variables (air temperature, relative humidity, and wind speed and direction) were measured by an automated weather station with a data logger that was installed in a chosen site to represent the two study areas. The chosen site is located approximately between the two study areas, and the station operated from 1 January to 31 December 2013.

### 2.3. Sampling

Twenty sampling sites were selected for the collection of surficial mangrove sediments (at a depth of 0–15 cm) and surface seawater samples in August 2013 from both the Sharam area and Alkhor area (Figure 1). At each site, samples were collected from three points and then pooled and kept at 5 °C for lab analysis. The seawater samples were collected in 500 mL plastic bottles. The sediment samples were collected using a stainless-steel auger and kept in zip-lock polyethylene bags. The collected sediment samples were air-dried at ambient temperature, and then the detritus and coarse materials contained in the sediment samples were discarded. Then, the sediment was sieved through a 0.075 mm mesh sieve, hand milled, and homogenized in a ceramic mortar.

### 2.4. PAH Analysis

PAHs were extracted from sediment samples using a Thermo Scientific dionex ASE^™^ 350 accelerated solvent extractor. Sediment samples of 5 g and diatomaceous earth (Sigma Aldrich^®^, St. Louis, MI, United States) were mixed and used to fill extraction cells of 34 mL. A dichloromethane/acetone mixture (1:1, *v*/*v*) at 100 °C and 1500 psi, three 5 min static extraction cycles and a flush volume of 60% were used to perform extraction. Under a steady nitrogen stream using a centrifugal evaporator (Genevac EZ-2 Solvent Evaporator, Ipswich, United Kingdom), dryness of extracts was achieved, and then dry extracts were reformed in approximately 2 mL of dichloromethane. The concentrated extract was filtered through a 0.2 µm PTFE filter (Chromafil Xtra MV-20/25, Laval, QC, Canada) with a 10 mL syringe and then transferred to a GC vial for GC/MS analysis. For water samples, each sample was filtered to remove suspended matter, and extraction was conducted following the USEPA Method 550.1 of Bashe and Baker in 1990 [23]. Using a C18 Empore^TM^ Solid Phase extraction disk (Neuss, Germany), 100 mL of each water sample was processed in a vacuum flask with sidearm for approximately 20 min. Then, the content of the disk was dried, and were eluted using 30 mL of dichloromethane twice. Under a steady nitrogen stream using a centrifugal evaporator (Genevac EZ-2 Solvent Evaporator, Ipswich, QLD, United Kingdom), the extract was concentrated to approximately 2 mL. Then, the concentrated extract was filtered through a 0.2 µm PTFE filter (Chromafil Xtra MV-20/25, Laval, Canada) with a syringe and transferred to a GC vial for GC/MS analysis. The GC/MS analysis was conducted with a JEOL JMS-GCmate System integrated with an HP6890 gas chromatograph. An Agilent J&W DB-EUPAH 20 m × 0.18 mm, 0.14 mm high efficiency GC column was used, and the following conditions were applied: oven temperature: 60 °C (1 min) → 260 °C (10 min) → 320 °C (4 min); injector temp: 280 °C; transfer line: 280 °C; ion source: 280 °C; analyzer: 150 °C; electron impact energy: 70 eV. A total of 1 µL volume of each sample was injected in the splitless mode, and the purge time was 1 min. Identification and quantification of 16 PAH compounds were based on matching retention times and their corresponding mass spectrum with a mixed-PAH standard (Dr. Ehrenstorfer GmbH L 20950009AL, PAH-Mix 9, Augsburg, Germany). The 16 identified PAH compounds were naphthalene (here abbreviated as Naph), acenaphthylene (Acy), acenaphthene (Ace), fluorene (Flu), phenanthrene (Phen), anthracene (Anth), fluoranthene (Flt), pyrene (Pyr), benzo[a]anthracene (BaA), chrysene (Chr), benzo[b]fluoroanthene (BbF), benzo[k]fluoroanthene (BkF), benzo[a]pyrene (BaP), dibenzo[a,h]anthracene (DahA), benzo[ghi]perylene (BghiP), and indeno[123-cd]pyrene (IcdP). All samples were run in selected ion monitoring (SIM) mode to enhance the sensitivity.

### 2.5. Degree of Similarity/Dissimilarity

The divergence degree of two datasets is determined using the divergence ratio (CD). To determine whether the measured PAHs at the investigated sites shared the same or different contaminating sources, the degree of discrepancy of the PAH contamination among the different sites within both investigated areas was calculated using the following CD [24].
(1)$$\mathrm{CDjk}=\sqrt{\frac{1}{p}\sum_{i=1}^{p}{\left(\frac{\mathrm{xij}-\mathrm{xik}}{\mathrm{xij}+\mathrm{xik}}\right)}^{2}}$$
where xij is the measured concentration of PAH contaminant i at a certain site, j and k are contaminant compared between sites, and *p* is the number of PAH contaminants. As the calculated CD approaches, one measurement from both sites is considered different, whereas as CD approaches zero, measurements from the two sites are considered similar. CD values lower than 0.27 between a contaminant from two sites can be attributed toward similar sources [24].

### 2.6. Quality Control Procedures

An external reference standard and procedural blanks were used as quality control methods. Dr. Ehrenstorfer GmbH L 20950009AL, PAH-Mix 9 was used as an external standard containing 16 PAH compounds for calibration and spiking the matrix. The matrix spike solutions were prepared from the mixed stock standards by volumetric dilution. Two procedural blanks were used for every 10-sample batch and processed together with the samples through the whole sample preparation and instrumental analysis. Limits of detection (LODs) were estimated as three times the standard deviation of the signal of the blanks. Bias in the sample matrix was estimated by adding target analytes at known concentrations to sample aliquots.

### 2.7. PAH Diagnostic Ratios

There are several methods, including molecular diagnostic ratios (MDRs), the principal component analysis (PCA) method [25,26], the chemical material balance (CMB) model [27], the positive matrix factorization (PMF) method, and stable carbon isotopic ratio analysis [28], used to identify sources of PAHs. In this study, MDRs were used to infer the possible sources of PAHs. Therefore, isomer pairs of PAHs, such as PAH of MW 202 Flt/(Flt + Pyr), 228 BaA/(BaA + Chr), 276 IcdP/(IcdP + BghiP) and low to high molecular weights of PAHs (L/H MW PAHs), were computed. Table 1 shows the diagnostic ratios utilized in this study along with value ranges reported for different sources.

## 3. Results and Discussion

### 3.1. Meteorology Measurements

Figure A1 shows how wind speed and direction were distributed in the study areas, and Table A1 shows the variation in the air temperature, relative humidity, and wind speed observed throughout the year. The hourly air temperature and relative humidity varied from 15 to 46 °C with a mean of 28.9 °C and from 2 to 93% with a mean of 52%, respectively. The hourly wind speed varied from approximately 0.0 ms^−1^ to 7.2 ms^−1^ with a mean of 2.1 ms^−1^. The predominant wind directions over the study areas were north-northwesterly (25.1%), followed by northwesterly (22.6%) and westerly (11.5%), with wind speeds predominantly occurring in the 1.37–3.06 category ms^−1^ (Table A2).

### 3.2. Concentrations and Composition of PAHs

The concentrations of 16 PAHs in the surface seawater and sediments of the study areas are summarized in Table 2. Generally, all 16 PAHs were detected at low concentrations throughout this study. Among 16 individual PAHs, Naph (2-ring PAH) had the highest mean concentration in sediment samples from the Sharam and Alkhor areas, 7.49 ng/kg and 0.02 ng/kg, respectively. For seawater samples, Phen (3-ring PAH) recorded the highest mean concentration (1.06 ng/L) in the Sharam area, and Pyr (4-ring PAH) recorded the highest mean concentration (3.25 ng/L) in the Alkhor area. The mean concentration of the 16 PAHs in the mangrove sediments in the Alkhor area (6.51 ng/kg) is relatively lower than that observed in the Sharam area (22.09 ng/kg), whereas in the mangrove seawater, it is higher in the Alkhor area (9.19 ng/L) than that observed in the Sharam area (4.33 ng/L). The frequency distribution for the concentration of different PAH compounds in all sediment and seawater samples of the two investigated areas is illustrated in Figure 2. From this figure, it can be observed that Phen and Pyr were the most dominant compounds in both sediment and seawater samples. Among the 16 analyzed PAHs, Flt, Phen, and Pyr had the highest detection frequency (100%) in sediment samples, whereas Flu, Phen, and Pyr had the highest frequency of occurrence in seawater samples. Anth, BbF, and BkF were detected in seawater samples with a detection frequency (30%) or below, whereas those PAH species were never detected in sediment samples. This concentration difference between the sediments and the column of seawater above them could be suggestive of older PAH contamination experience in the Sharam area than the likely recent contamination experience in the Alkhor area.

The composition pattern of PAHs detected in the samples by the number of rings is shown in Figure 3. The sediment of all 10 sites as well as seawater of four sites (3, 4, 5, and 7) in the Sharam area featured a higher abundance of 2,3-ring PAHs. Similarly, in the Alkhor area, the sediment of all sites except site 6 as well as the seawater of two sites (5 and 6) featured a higher abundance of 2,3-ring PAHs. HMW-PAHs were generally predominant in sediment sites compared to LMW-PAHs. This predominance of LMW PAHs may be attributed to preferential degradation during PAH transport and burial into sediments [29]. Commonly, a higher abundance/concentration of HMW PAHs compared to that of LMW PAHs is observed in sediments from river and marine environments (e.g., [30]).

On the basis of molecular weight, comparisons of low-molecular-weight (LMW) PAH (MW < 228) and high-molecular-weight (HMW) PAH (MW >228) of sediment samples in both study areas were conducted: 11.00 compared to 11.09 ng/kg in the Sharam area and 3.26 compared to 3.25 ng/kg in the Alkhor area, respectively. It was found that LMW PAH compounds composed of fewer than four aromatic rings were dominant in sediment samples collected from both study areas (70% in the Sharam area and 66% in the Alkhor area). A close level of abundance of fewer-than-4-ring PAHs (44%) and more-than-4-ring PAHs (40%) and to some degree the relative proximity in LMW and HMW PAH concentrations (1.59 and 2.73 ng/L, respectively) were detected in Sharam area water. However, it was found that seawater samples from the Alkhor area had a close level of abundance of 4-ring PAHs (40%) and fewer-than-4-ring PAHs (38%). As a result, it was dominated by HMW PAHs (7.98 ng/L), as shown in Table 2 and Figure 3. The observed difference between the HMW PAH abundance in seawater and sediment samples of both investigated areas may indicate significant HMW PAH modification by water column processes during sedimentation.

### 3.3. Dissimilarity between the PAH Contamination in the Two Areas

The dissimilarity matrix among the sampled sites of both investigated areas is shown in Figure 4 for sediment and Figure 5 for seawater samples. A degree of dissimilarity ranging from 0.315 to 0.860 and from 0.293 to 0.692 for sediment and seawater, respectively, was observed among all sites (Figure 4 and Figure 5). Therefore, the two areas do not share common sources of PAHs and are not impacted by long-range pollution transport; rather, they are impacted by site-specific PAH contamination sources. According to the prevalent wind direction (NNW as shown in Table A2), the Sharm area is susceptible to emissions released from the desalination plant located northwest of the Sharm area. On the other hand, the Alkhor area is not impacted by this source of contamination since the wind coming from the south and south-south-west are less than 1.5% as shown in Table A2. Differences among the PAH-contaminated sediment samples of the Sharam area were more pronounced than those of the Alkhor area (Figure 4), whereas the results in seawater were reversed (Figure 5). This result suggests that the site-specific contamination sources have a relatively greater distinct impact on Sharam sediment than on Alkhor sediment, whereas they have a relatively greater distinct impact on Alkhor seawater than on Sharam seawater. However, the two study areas are features low PAHs compared to other studies in the literature (Table A3).

### 3.4. Potential Sources of PAHs Pollution

The possible PAH sources using selected MDRs, namely, PAHs (L/H MW), Flt/(Flt + Pyr), IcdP/(IcdP  +  Bghip), and BaA/(BaA  +  Chr), are illustrated in Figure 6. From Figure 6a, it can be observed that the majority of PAHs in sediment samples of both study areas originated from petrogenic sources, while minor amounts originated from pyrogenic sources. However, the majority of PAHs in seawater samples originated from pyrogenic sources associated with few petrogenic sources. From the bivariate plot for IcdP/(IcdP  +  Bghip) versus Flt/(Flt + Py) ratios (Figure 6b), it can be observed that the PAHs of sediment samples for both study areas originate from liquid fossil fuel combustion sources associated with a minor fraction from unburned petroleum sources. Seawater samples mainly originated from mixed sources of liquid fossil fuel, biomass, and coal combustion. Furthermore, the PAH cross plot for BaA/(BaA + Chr) versus Flt/(Flt + Pyr) ratios (Figure 6c) of sediment samples of both study areas indicates mixed sources of petroleum and petroleum combustion sources. Seawater samples originate from mixed sources of liquid fossil fuel, biomass, and coal combustion. These cross plots showed that sediment samples for both investigated areas originated from petroleum and petroleum combustion, in particular, liquid fuel combustion. Seawater samples originated from coal and biomass combustion.

## 4. Conclusions

The findings from this work represent vital reference information for future development and risk assessment studies in the Red Sea coastal area. This study offers important information on the contamination levels and potential origins of 16 priority PAHs in mangrove seawater and sediments in two investigated areas located along the Red Sea Coast. The present baseline measurements of PAH contaminants in the two investigated coastal areas would serve as a useful tool for future assessment of the ecosystem in the coastal sea area. This study revealed that the average concentration of total PAH mangrove sediments in the Sharam area is much larger than that in the Alkhor area, while the average concentration of the total PAH in the mangrove seawater in the Alkhor area is double that in the Sharam area. Additionally, the PAHs in sediment samples and seawater samples of both study areas were attributed mainly to petrogenic sources and pyrogenic sources, respectively. Mangrove exposure to PAHs due to the enhanced levels of motorboat usage and discharge of wastewater from nearby industrial sites and service locations to seawater bodies in coastal sea areas are of high concern and should be monitored and controlled. Periodic measurement of PAH concentrations at intervals of 8–10 years is recommended to maintain a healthy aquatic ecosystem.

## Figures and Tables

**Figure 1 ijerph-19-05474-f001:**
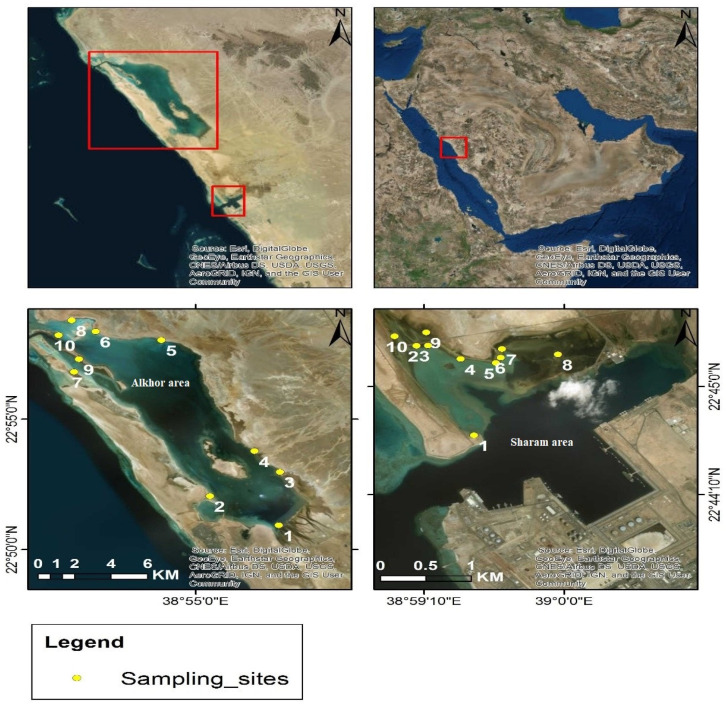
Map of the study areas (Alkhor area and Sharam area) showing the locations of sample collection sites.

**Figure 2 ijerph-19-05474-f002:**
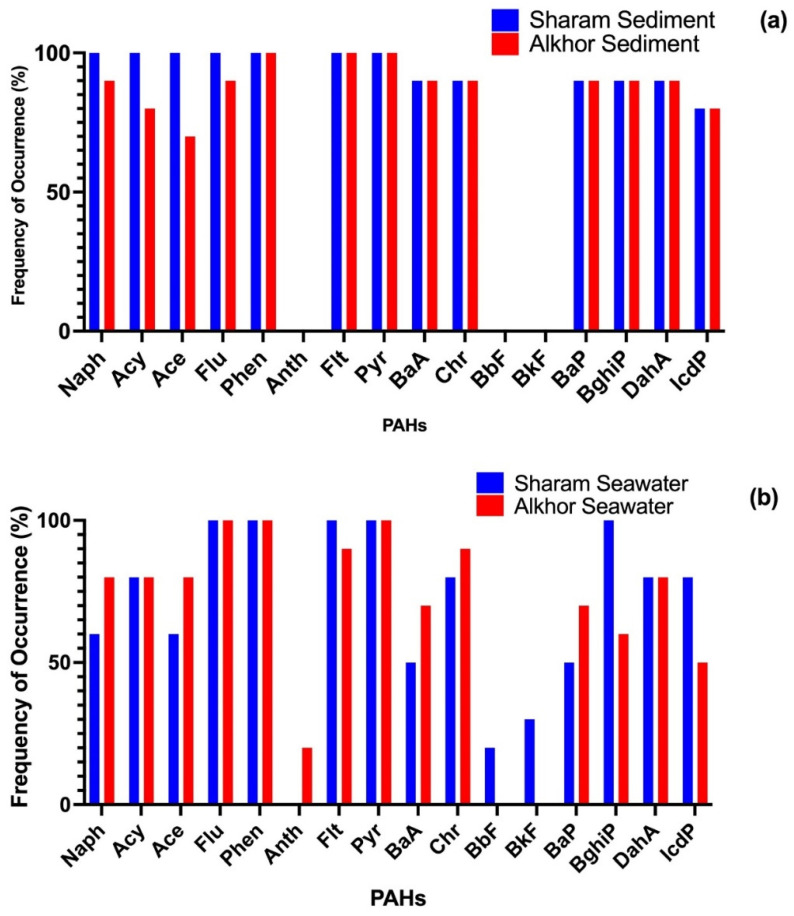
Frequency of occurrence (%) of PAHs in sediment and water samples from the Sharam area and Alkhor area. (**a**) represents sediment samples, (**b**) represents seawater samples.

**Figure 3 ijerph-19-05474-f003:**
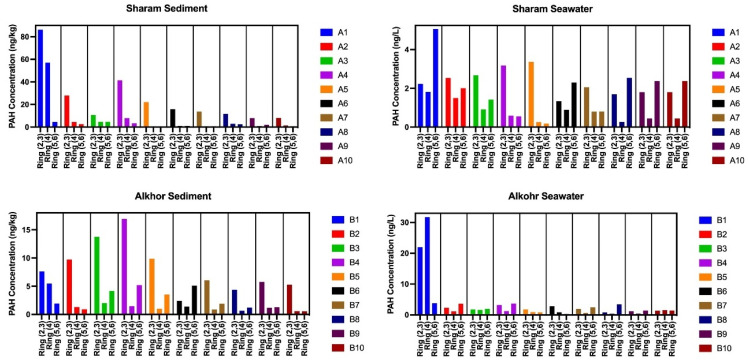
Concentrations of PAHs with different numbers of rings in the water and sediment samples from the Sharam area and Alkhor area.

**Figure 4 ijerph-19-05474-f004:**
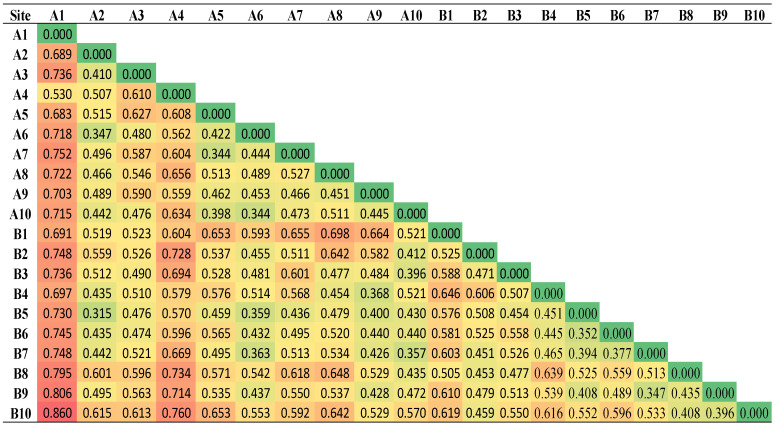
The dissimilarity matrix among the sampled sites of the sediment in both investigated areas.

**Figure 5 ijerph-19-05474-f005:**
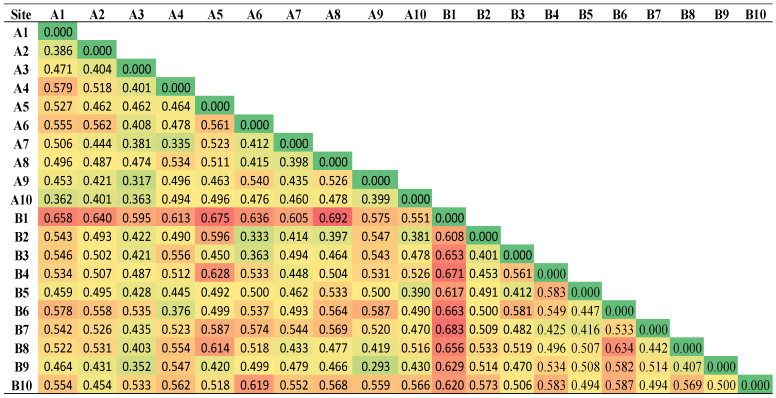
The dissimilarity matrix among the sampled sites of the seawater in both investigated areas.

**Figure 6 ijerph-19-05474-f006:**
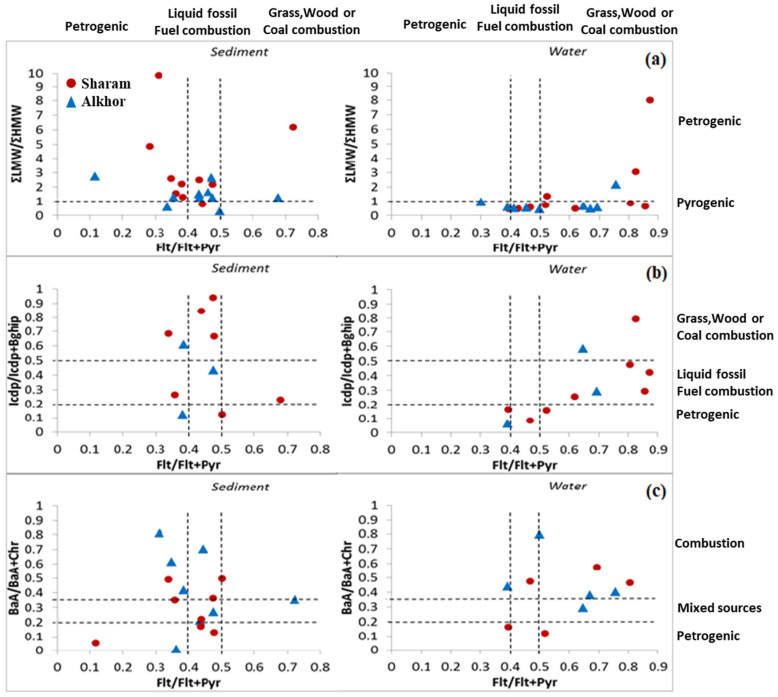
PAH cross plots of the diagnostic ratios used for both study areas: (**a**) the ratio PAH (L/H MW) vs. Flt/(Flt + Pyr), (**b**). the ratio IcdP/(IcdP  +  Bghip) vs. Flt/(Flt + Pyr), and (**c**) the ratio BaA/(BaA  +  Chr) vs. Flt/(Flt + Pyr).

**Table 1 ijerph-19-05474-t001:** PAH diagnostic ratios employed in this study.

PAH Ratio	Value Ranges	Source
**PAHs (L/H MW)**	>1 <1	Petrogenic Pyrogenic
**Flt/(Flt + Pyr)**	<0.4 0.4–0.5 >0.5	Petrogenic Fossil fuel combustion grass/wood/coal combustion
**BaA/(BaA + Chr)**	<0.2 0.2–0.35 >0.35	Petrogenic Mixed sources Combustion
**IcdP/(IcdP** **+** **BghiP)**	<0.2 0.2–0.35 >0.35	Petrogenic Mixed sources Combustion

Flt: Fluoranthene/C_16_H_10_; Pyr: Pyrene/C_16_H_10_; BaA: Benzo[a]anthracene/C_18_H_12_; Chr: Chrysene/C_18_H_12_; IcdP: Indeno[1,2,3-cd]pyrene/C_22_H_12_; Bghip: Benzo[ghi]perylene/C_22_H_12_.

**Table 2 ijerph-19-05474-t002:** Concentrations of PAHs in the water and sediment samples from the Sharam area and Alkhor area.

PAH	# Rings	Sharam Area				Alkhor Area			
		Sediment		Seawater		Sediment		Seawater	
		Mean (ng/kg)	Std. dev.	Mean (ng/L)	Std. dev.	Mean (ng/kg)	Std. dev.	Mean (ng/L)	Std. dev.
**Naph**	2	7.49	7.53	0.14	0.16	2.02	1.32	0.23	0.33
**Acy**	3	0.23	0.21	0.09	0.08	0.10	0.10	0.10	0.08
**Ace**	3	0.65	0.87	0.08	0.08	0.12	0.16	0.19	0.16
**Flu**	3	0.44	0.25	0.22	0.10	0.20	0.11	0.30	0.19
**Phen**	3	2.18	2.09	1.06	0.32	0.82	0.30	0.36	0.18
**Anth**	3	N/A	N/A	N/A	N/A	N/A	N/A	0.03	0.08
**Flt**	4	3.17	7.75	0.40	0.29	0.36	0.50	2.29	5.88
**Pyr**	4	5.07	12.46	0.20	0.12	0.53	1.00	3.25	9.26
**BaA**	4	0.18	0.18	0.10	0.18	0.10	0.11	0.22	0.22
**Chr**	4	0.36	0.51	0.23	0.24	0.30	0.18	0.25	0.24
**BbF**	5	N/A	N/A	0.07	0.16	N/A	N/A	N/A	N/A
**BKF**	5	N/A	N/A	0.09	0.19	N/A	N/A	N/A	N/A
**BaP**	5	0.75	0.82	0.20	0.25	0.38	0.41	0.33	0.36
**DahA**	5	0.62	0.43	0.55	0.42	1.01	0.85	0.75	0.59
**IcdP**	6	0.41	0.68	0.18	0.15	0.26	0.18	0.23	0.29
**BghiP**	6	0.51	0.49	0.71	0.49	0.31	0.48	0.66	1.02
**ΣPAHs**		22.09		4.33		6.51		9.19	
**ΣLMW**		11.00		1.59		3.26		1.20	
**ΣHMW**		11.09		2.73		3.25		7.98	

## Data Availability

Not applicable.

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
