# Peer review of "Contamination Assessment of Mangrove Ecosystems in the Red Sea Coast by Polycyclic Aromatic Hydrocarbons"

_ijerph, 2022, doi:10.3390/ijerph19095474_

Round 1

Reviewer 1 Report

The manuscript entitled “Contamination Assessment of Mangrove Ecosystems in the Red 2 Sea Coast by Polycyclic Aromatic Hydrocarbons” is well written. I suggest some of the modifications below,

  1. What is the rationale for the selection of meteorological variables
  2. What is the basis for the selection of Twenty sampling sites
  3. Discussion the manuscript is very poor, please improve with appropriate literature
  4. Most of the results are relevant to actual meteorological variables, therefore please associate the discussion with cited literature

Author Response

The manuscript entitled “Contamination Assessment of Mangrove Ecosystems in the Red 2 Sea Coast by Polycyclic Aromatic Hydrocarbons” is well written. I suggest some of the modifications below,

  1. What is the rationale for the selection of meteorological variables

Response 1:

The rational is that this study is a baseline measurements and the meteorological variables are important for such study to capture the predominant atmospheric conditions in the targeted areas. 

______________________________________________________________________________

  1. What is the basis for the selection of Twenty sampling sites

Response 2:

 Systematic sampling technique was used in the targeted mangrove areas.

______________________________________________________________________________

  1. Discussion the manuscript is very poor, please improve with appropriate literature

Response 3:

We have added the following table and discussions to fulfill the requirements raised by the reviewer.

A table was added in line 291 as a supplementary data (Table A3)

 Table A3. The Concentrations of PAHs in selected studies.

Study

Location

Concentrations

Number of PAH

This study

Rabigh, Saudi Arabia

<0.001 - 0.073 ng/g

16 PAH

Viguri, J., et al. (2002)

Santander Bay, Spain

0.02 - 344.6 μg/g

16 PAH

McCready, S., et al. (2000)

Sydney, Australia

<100 - 380 000 ug/kg

16 PAH

Dong, Cheng-Di, et al., (2012)

Taiwan

4425 - 51261 ng/g

16 PAH

Qiao, Min, et al. (2006)

Taihu Lake, China

1207 - 4754 ng/g

16 PAH

Adami, Gianpiero, et al. (2000)

Trieste, Adriatic Sea

2.37 - 64.56 ug/g

16 PAH

Oen, Amy MP, et al., (2006): 370-380.

Norway

2 - 113 ug/g

16 PAH

De Luca, Giuseppe, et al. (2004)

Sardinia, Italy

0.07 - 1.21 ug/g

16 PAH

Simpson, Christopher D., et al. (1996)

Kitimat, Canada

<1 - 10000 ug/g

16 PAH

Witt, G. (1995)

Baltic Sea

9 - 29 ng/g

15 PAH (sandy areas)

Witt, G. (1995)

Baltic Sea

800 - 1900 ng/g

15 PAH (sediment)

Viguri, J., J. Verde, and A. Irabien. "Environmental assessment of polycyclic aromatic hydrocarbons (PAHs) in surface sediments of the Santander Bay, Northern Spain." Chemosphere 48.2 (2002): 157-165.

McCready, S., et al. "The distribution of polycyclic aromatic hydrocarbons in surficial sediments of Sydney Harbour, Australia." Marine pollution bulletin 40.11 (2000): 999-1006.

Dong, Cheng-Di, Chih-Feng Chen, and Chiu-Wen Chen. "Determination of polycyclic aromatic hydrocarbons in industrial harbor sediments by GC-MS." International journal of environmental research and public health 9.6 (2012): 2175-2188.

Qiao, Min, et al. "Composition, sources, and potential toxicological significance of PAHs in the surface sediments of the Meiliang Bay, Taihu Lake, China." Environment International 32.1 (2006): 28-33.

Adami, Gianpiero, et al. "Detecting and characterising sources of persistent organic pollutants (PAHs and PCBs) in surface sediments of an industrialized area (harbour of Trieste, northern Adriatic Sea)." Journal of Environmental Monitoring 2.3 (2000): 261-265.

Oen, Amy MP, Gerard Cornelissen, and Gijs D. Breedveld. "Relation between PAH and black carbon contents in size fractions of Norwegian harbor sediments." Environmental Pollution 141.2 (2006): 370-380.

De Luca, Giuseppe, et al. "Polycyclic aromatic hydrocarbons assessment in the sediments of the Porto Torres Harbor (Northern Sardinia, Italy)." Marine chemistry 86.1 (2004): 15-32.

Simpson, Christopher D., et al. "Composition and distribution of polycyclic aromatic hydrocarbon contamination in surficial marine sediments from Kitimat Harbor, Canada." Science of the Total Environment 181.3 (1996): 265-278.

Witt, G. "Polycyclic aromatic hydrocarbons in water and sediment of the Baltic Sea." Marine Pollution Bulletin 31.4 (1995): 237-248.

A sentence was added in line 238 as follows:

However, the two study areas are features low PAHs compared to other studies in the literature [Table A3].

______________________________________________________________________________

  1. Most of the results are relevant to actual meteorological variables, therefore please associate the discussion with cited literature

Response 4:

We have added the following two sentences to the manuscript:

“According to the prevalent wind direction (NNW as shown in table A2), the Sharma area is susceptible to emissions released from the desalination plant located northwest of Sharma area. On the other hand, the Alkhor area is not impacted by this source of contamination since the wind coming from south and south-south-west are less than 1.5% as shown in table A2”.

Reviewer 2 Report

The manuscript submitted by Prof.  Badr H. Alharbi and coworkers is interesting and can be accepted after some minor modifications. Some specific suggestions are given below:

1) The first sentence itself is lengthy. It could be compressed and presented as two sentences.

2) In many places "mL" is wrongly given as "ml". This should be corrected wherever applicable.

3) Line 101, "Then, the disk was dried, and.....". This text should be rephrased to make it clear.

4) Line 111, Delete "A" from "A 1 µL volume....".

5) Change "ng/l" to more appropriate "ng/L" throughout the manuscript.

6) Table 2 will be more clear if the abbreviations are explained as footnote.

7) Line 252, Put comma before and after "in particular".

The manuscript may be published after addressing these comments.

Author Response

The manuscript submitted by Prof.  Badr H. Alharbi and coworkers is interesting and can be accepted after some minor modifications. Some specific suggestions are given below:

The manuscript may be published after addressing these comments.

1) The first sentence itself is lengthy. It could be compressed and presented as two sentences.

Response 5:

As requested by the reviewer, we have separated the first sentence in the introduction into two sentences to be as follows in page 1 line 27-28:

“Globally, approximately 50% of mangroves have been lost over recent decades. The main human activities for losing mangroves are reclamation, farming, aquaculture, deforestation, and urban development”

______________________________________________________________________________

2) In many places "mL" is wrongly given as "ml". This should be corrected wherever applicable.

Response 6:

As a response to the reviewer’s comment, we have changed all the ml in the manuscript into mL.

We corrected 7 words in line 82, 92, 96, 98, 101, 103, and 105.

_____________________________________________________________________________

3) Line 101, "Then, the disk was dried, and.....". This text should be rephrased to make it clear.

Response 7:

The sentence was corrected to be as follows:

“Then, the content of the disk was dried, and were eluted using 30 mL of dichloromethane twice.”

______________________________________________________________________________

4) Line 111, Delete "A" from "A 1 µL volume....".

Response 8:

The letter “A” was removed to make the sentence as follows:

“1 µL volume of each sample was injected in the splitless mode, “

______________________________________________________________________________

5) Change "ng/l" to more appropriate "ng/L" throughout the manuscript.

Response 9:

As requested by the reviewer, it was corrected in the whole manuscript. ng/l was changed to ng/L.

The changes were occurred in the following lines: 16, 17, 171, 172, 175, 205, and 208.

______________________________________________________________________________

6) Table 2 will be more clear if the abbreviations are explained as footnote.

Response 10:

As requested, the abbreviations of the table’s content were written as footnote as follows:

Flt: Fluoranthene/C16H10; Pyr: Pyrene/C16H10; BaA: Benzo[a]anthracene/C18H12; Chr: Chrysene/C18H12; IcdP: Indeno[1,2,3-cd]pyrene/C22H12; Bghip: Benzo[ghi]perylene/C22H12

______________________________________________________________________________

7) Line 252, Put comma before and after "in particular".

Response 11:

As requested, a comma was added before and after the phrase “in particular.

Reviewer 3 Report

This is an interesting topic. The authors investigate the occurrence of PAHs in sediment and water from different two areas and get some interesting results. However, there are some flaws in this paper. I recommend a major revision.

Sampling

I see the author didn’t conduct the field blank. This is also important due to the potential pollution during sampling.

The author used air-dried to dry the sediment samples. I think the common method is freeze-drying for organic pollutants analysis in sediment due to the potential contamination during drying. Did the author conduct the dring procedure blank?

Results and discussion

Line 225-228: Differences among the PAH-contaminated sediment samples of the Sharam area were more pronounced than those of the Alkhor area (Table 3), whereas the results in seawater were reversed.

3.4 Potential sources of PAHs Pollution: The authors found that the sources of PAHs in water and sediments were different in the two areas. I'm a little confused about this. Why are the sources different, and has anyone dumped mud in these two areas? As far as I know, PAHs in sediments also come from water because of their hydrophobicity. I think the authors should combine the results of PAHs in water and sediments for MDRs analysis.

Author Response

This is an interesting topic. The authors investigate the occurrence of PAHs in sediment and water from different two areas and get some interesting results. However, there are some flaws in this paper. I recommend a major revision.

Sampling

I see the author didn’t conduct the field blank. This is also important due to the potential pollution during sampling.

 The author used air-dried to dry the sediment samples. I think the common method is freeze-drying for organic pollutants analysis in sediment due to the potential contamination during drying. Did the author conduct the dring procedure blank?

 Response 12:

Due to the fact that sediment certified as free of organic compounds was not available, no field blanks were used in this study. Moreover, no drying procedure blank was used. However, we followed best practices to avoid contamination of the samples by equipment and cross contamination between sites.

______________________________________________________________________________

Results and discussion

Line 225-228: Differences among the PAH-contaminated sediment samples of the Sharam area were more pronounced than those of the Alkhor area (Table 3), whereas the results in seawater were reversed.

 Response 13:

We have changed the text as suggested by the reviewer.

______________________________________________________________________________

3.4 Potential sources of PAHs Pollution: The authors found that the sources of PAHs in water and sediments were different in the two areas. I'm a little confused about this. Why are the sources different, and has anyone dumped mud in these two areas? As far as I know, PAHs in sediments also come from water because of their hydrophobicity. I think the authors should combine the results of PAHs in water and sediments for MDRs analysis.

 Response 14:

This is another insightful observation. As a matter of fact, both study areas used to receive massive annual runoffs until 2009 when a water dam was established. Therefore, it is quite possible that the observed different sources were attributed to natural mud dumping associated with runoffs. As for the MDRs analysis, it would be a valuable addition for the future periodic assessments as mentioned in the conclusion.

Round 2

Reviewer 3 Report

None